# Advanced Online Monitoring of In Vitro Human 3D Full-Thickness Skin Equivalents

**DOI:** 10.3390/pharmaceutics14071436

**Published:** 2022-07-08

**Authors:** Roland Schaller-Ammann, Sebastian Kreß, Jürgen Feiel, Gerd Schwagerle, Joachim Priedl, Thomas Birngruber, Cornelia Kasper, Dominik Egger

**Affiliations:** 1Health—Institute for Biomedicine and Health Sciences, Joanneum Research Forschungsgesellschaft mbH, Neue Stiftingtalstrasse 2, 8010 Graz, Austria; roland.schaller-ammann@joanneum.at (R.S.-A.); juergen.feiel@joanneum.at (J.F.); gerd.schwagerle@joanneum.at (G.S.); joachim.priedl@joanneum.at (J.P.); 2Institute of Cell and Tissue Culture Technologies, Department of Biotechnology, University of Natural, Resources and Life Sciences, Vienna, Muthgasse 18, 1190 Vienna, Austria; sebatian.kress@boku.ac.at (S.K.); cornelia.kasper@boku.ac.at (C.K.)

**Keywords:** open flow microperfusion, pharmacokinetics, penetration tests, skin explants, full-thickness skin equivalent, sensors, oxygen, pH, glucose, lactate

## Abstract

Skin equivalents and skin explants are widely used for dermal penetration studies in the pharmacological development of drugs. Environmental parameters, such as the incubation and culture conditions affect cellular responses and thus the relevance of the experimental outcome. However, available systems such as the Franz diffusion chamber, only measure in the receiving culture medium, rather than assessing the actual conditions for cells in the tissue. We developed a sampling design that combines open flow microperfusion (OFM) sampling technology for continuous concentration measurements directly in the tissue with microfluidic biosensors for online monitoring of culture parameters. We tested our design with real-time measurements of oxygen, glucose, lactate, and pH in full-thickness skin equivalent and skin explants. Furthermore, we compared dermal penetration for acyclovir, lidocaine, and diclofenac in skin equivalents and skin explants. We observed differences in oxygen, glucose, and drug concentrations in skin equivalents compared to the respective culture medium and to skin explants.

## 1. Introduction

Ethically responsible research and development aim for the replacement of animal experiments. Hence, already in 1959, the 3R principles were introduced in the scientific community to refine, reduce, and ultimately replace animal experiments [1]. Two-dimensional cell culture is a means to reduce animal experiments for example when potential drug candidates can be pre-screened for further evaluation. However, a 2D monoculture setting is highly simplified relative to the complexity of an organism [2]. Two-dimensional monocultures might produce false positive and false negative results when it comes to finding a specific drug for systemic application. Animal experiments are thus still inevitable for efficacy testing.

With the advent of tissue engineering which is using cells, cellular products, materials, and engineering approaches to create functional tissue substitutes [3] a wide variety of tissue equivalents has been developed, for, e.g., skin [4,5], liver [6], intestinal [7,8], cardiac [9], and lung tissue [10]. Originally intended to replace dysfunctional tissue to restore, maintain, or improve tissue functionality in a patient, tissue equivalents are also suited for in vitro test systems to study tissue functionality, intercellular interaction, cell-matrix reciprocity in physiological and pathological situations as well as mechanisms of, e.g., wound healing and drug uptake [11]. Those equivalents are set in a 3D environment with co-cultures of primary human cell types to study human physiology and cellular responses [12]. Moreover, equivalents can be established with patient-derived cells to generate personalized tissues for precision medicine [13,14].

Especially in the field of skin tissue engineering, highly advanced skin equivalents have been developed that closely mimic native skin, including pigmentation [15] and vascularization [16], and can be applied to investigate, e.g., immune reaction [17], burn wounds [18], and wound treatment [19]. Such in vitro equivalents also exhibit advantages over ex vivo experiments using skin explants from surgery. While skin explants exhibit donor variance and often originate from waste material that might encompass clinical conditions, the assembly of in vitro equivalents can be standardized resulting in a reproducible product and hence reliable and translatable results.

Despite the improvement of sophisticated tissue equivalents for the investigation of pathophysiological processes, the investigation of the respective 3D equivalents constitutes the current limitation to replace animal models and exploit the full potential of tissue equivalents. Especially for complex 3D tissue equivalents, it is crucial to develop minimally invasive non-destructive analytical methods to facilitate continuous or repetitive analysis and monitoring of tissue viability, development, functionality, and cellular responses reliably, reproducibly, and preferably in real time [20,21].

There are commercially available systems to study the in vitro penetration of drugs and active substances to conclude on dosage forms. The Franz diffusion chamber is one of the most important methods for determining transdermal drug administration [22] in skin explants or in vitro generated skin equivalents. However, this allows only monitoring the concentration of the test substance in a receptor medium rather than giving information on the actual concentration and condition inside the tissue. This setup informs about the permeability and barrier function of the inserted skin rather than the pharmacodynamics and pharmacokinetics in the skin. Further methods such as transepithelial electrical resistance measurements also indicate, e.g., wound healing [21,23] but also only on the basis of the barrier integrity not indicating the actual tissue condition. Another attempt to monitor the topical condition of wounds is the application of wound dressings containing, e.g., pH indicators [24].

A change in traditional culture parameters, such as oxygen, pH, glucose, or lactate can have detrimental effects on cell behavior and thus on the relevance of the generated data [25]. These parameters are currently only measured in the culture medium although we have recently demonstrated that the actual values of these parameters in 3D-cultured in vitro equivalents differ significantly from the values measured in the culture medium [26]. In skin equivalents or skin explants used for pharmacological studies, these parameters are rarely assessed while only reliable monitoring of these parameters within the tissue will enable a tight control of experimental conditions.

The aim of the study was to develop a novel ex vivo incubator in combination with minimally-invasive open flow microperfusion (OFM) sampling technology [27], and an integrated microfluidic biosensor platform [26] for the online monitoring of critical cell culture parameters. To validate this advanced online monitoring, we used real-time assessment of oxygen, glucose, lactate, and pH in the dermis of an in vitro 3D full-thickness skin equivalent (FTSE) and the respective culture medium. Furthermore, we performed penetration studies with acyclovir, diclofenac, and lidocaine in the FTSE and compared it with penetration in skin explants.

## 2. Materials and Methods

### 2.1. Ex Vivo Incubator

For penetration experiments, the OECD guideline [28] stipulates that the design of the diffusion cells known from the literature [29,30,31] must allow precise temperature control of the skin and the receptor fluid. Furthermore, the surface temperature of skin explants should be 32 ± 1 °C during the penetration test, corresponding to the surface temperature of living human skin. Changes in skin temperature can affect the absorption process since diffusion is temperature dependent [32]. As a result, incorrect penetration behavior would be determined by superficially applied creams and must therefore be prevented.

It is also important that moisturizing prevents the skin from drying out. If the environment is too humid, the release rate will be too high, or if the environment is too dry, it will be reduced. Therefore, similar to testing medicines in the clinic, the humidity must be kept constant between 40 and 60% [33].

Since no suitable device is commercially available, a mobile “ex vivo incubator” with three sliding doors for penetration tests with human skin explants or FTSE was developed for this study (Figure 1). The technical requirements were: a temperature control range of 30–38 °C, a humidity control range of 40–60%, smooth, easy to wipe, and alcohol-resistant surfaces as well as no dead zones. The dimensions of the ex vivo incubator were: 140 × 86 × 75 cm (L × W × H) with a maximum weight of the laminar box without components of 80 kg to keep it mobile using a trolley.

The evaluation of the ex vivo incubator showed a run-in phase of approximately 50 min after which the temperature was regulated with an evaluated accuracy of 32.2 ± 0.67 °C and humidity of 53.2 ± 2.42%. The system comprises a modified climate control unit (Trixie, CCU), two heaters (Sichler, Type: NC-6469-675d), two humidifiers (Trixie, Fogger XL), and a ventilator (Dark Force, Black Fan). Both heaters were covered with Styrofoam^TM^ to prevent direct heat radiation and thus prevent the skin/ FTSE from drying out.

### 2.2. OFM Technology

OFM is a sampling technology for clinical and preclinical drug development studies and biomarker research. OFM was designed for continuous sampling of analytes from the interstitial fluid (ISF) of various tissues (e.g., skin, adipose tissue, brain). It provides direct access to the ISF by insertion of a small, minimally invasive, membrane-free OFM probe with macroscopic openings [27]. OFM is capable of sampling lipophilic and hydrophilic compounds [34], protein-bound [35,36] and unbound drugs, neurotransmitters, peptides and proteins, antibodies [37,38,39], nanoparticles and nanocarriers, enzymes, and vesicles. Thus, OFM sampling accesses the entire biochemical information of the ISF regardless of the analyte’s molecular size, protein-binding property, or lipophilicity. For sampling, linear OFM probes (a/dOFM-P-15, Joanneum Research, Graz, Austria) were implanted at a depth of approximately 1 mm into the dermis of skin explants (Figure 2A) and skin equivalents (Figure 2B) using sterile cannulae (Sterican 0.90 × 70 mm, B. Braun, Melsungen, Germany). In case of skin explants, entry and exit points were sealed using cyanoacrylate adhesive (732, Panacol, Steinbach, Germany). The OFM probes were operated using OFM tubings (SCS001), OFM bags (PEB001), and OFM pumps (MPP102 PC, all Joanneum Research, Austria) in push-pull-mode at flow rates of 1 µL/min. Flow rates were verified by calculation, taking the extracted volume of ISF/perfusate (OFM sample) and sampling interval into consideration. The perfusate used to operate the OFM probes consisted of a physiological electrolyte solution for medical infusion purposes (ELO-MEL Isoton, Fresenius Kabi, Graz, Austria) with an added concentration of 1% human serum albumin (Albunorm, Octapharma GmbH, Heidelberg, Germany).

### 2.3. Online Monitoring

Online monitoring of oxygen, pH, glucose, and lactate was performed by connecting two individual flow-through cells consecutively after the outlet of the OFM probes (Figure 2C and Figure 3). Flow-through cells were optimized to enable reasonable response times, accuracy, and precision at a flow rate of 1 µL/min, when being temperature compensated. OFM probes manufactured from PEEK (polyetheretherketone) are impermeable for oxygen and thus allow accurate quantification from OFM samples.

The lead flow-through cell from PyroScience held optical sensor spots for oxygen and pH measurements and featured a minimized internal cell volume of less than 10 µL. In parallel, the same optical sensor spots were used to measure oxygen and pH in the Petri dish that contained culture medium to cultivate the FTSE (Figure 2). All sensor spots were connected via optical fibers to a 4-channel read-out instrument (FireSting^®^-PRO). Oxygen sensors showed a dynamic range from 0 to 100% air saturation, whereas pH sensors showed a range of 6 to 9. Data acquisition was performed using “PyroWorkbench” (all by PyroScience, Aachen, Germany).

For online monitoring of glucose and lactate, a second flow-through cell from Jobst (LV5, Jobst Technologies, Breisgau, Germany) with tubing of 0.5 mm inner diameter was connected to the outlet of the PyroScience flow-through cell. The Jobst flow-through cell had to be re-evaluated for a flow rate of 1 µL/min because the manufacturer’s specifications only cover flow rates down to 5 µL/min. Successful re-evaluation showed suitable stability, accuracy, precision, and an acceptable response time (data not shown). The glucose sensor showed a range of <0.9 to 450 mg/dL, whereas the lactate sensor showed <0.18 to 135 mg/dL. Data acquisition was performed using the biosensor measuring instrumentation (Six) and software (bioMON, v4.15.0, all from Jobst Technologies).

The oxygen sensor was referenced using air-saturated water (100%) or oxygen-free water (0%) by using Oxcal (PyroScience, Aachen, Germany). The pH sensor was referenced using calibration capsules pH 2 and pH 10 (both from PyroScience, Germany). Glucose and lactate sensors were tested individually, before and after the investigations, using 3 individual referenced stock solutions per analyte within the operation range, to determine drift, sensitivity loss, or sensor damage. For retrospective analysis, false values caused by air bubbles were excluded and linearly interpolated. Manual samples of the culture medium were taken frequently and analyzed offline for pH, glucose, and lactate. The removed volume was replaced, and unintentional evaporation was compensated with distilled water.

Online readings of glucose and lactate in OFM samples from FTSE tissue and skin explants were referenced using a pre-evaluated analyzer (data not shown, Super GL2, Dr. Müller Gerätebau, Freital, Germany). Offline pH values were estimated using test strips (Dosatest, VWR chemical, Darmstadt, Germany).

### 2.4. Full Thickness SKIN Equivalent (FTSE)

Phenion^®^ Full-Thickness LARGE Skin Models (Henkel, Düsseldorf, Germany) with a diameter of 3.1 cm each were used as FTSE. This FTSE is a multilayer equivalent to human skin and consists of human keratinocytes and fibroblasts populated on a collagen matrix. For this study, the FTSE was cultivated in sterile conditions at Henkel Laboratories for an additional week, in order to increase the thickness of the stratum corneum and thus increase the penetration barrier (confirmed by histological investigations, data not shown).

FTSEs were delivered on transport agar in well plates. An O-shaped patch (Cutimed Hydro L, BSN medical GmbH, Hamburg, Germany) with an outer diameter of 3.1 and an inner diameter of 1.8 cm was bonded onto the FTSE surface to form round application sites (2.54 cm^2^) and prevent cross-contamination of the culture medium (Air Liquid Interface Culture medium, ALI CM-250, Henkel, Germany). OFM probes were implanted according to the manufacturer’s instructions and implanted FTSEs were placed into Petri dishes (VWR, Darmstadt, Germany) inside the ex vivo incubator with integrated sensor spots for oxygen and pH, on top of spacers and filter paper. Petri dishes were filled with culture medium and placed in sampling units designed and manufactured by Joanneum Research. The CAD models for the sampling units were generated with SolidWorks^®^ (Dassault Systems SolidWorks Corp., Waltham, MA, USA), 3D printed in clear resin using a Form 2 (both FormLabs, Sommerville, MA, USA) SLA 3D desktop printer. Then, implanted OFM probes were connected to OFM pumps, tubing, an online monitoring system, and sampling vials. FTSE online monitoring was performed in triplicates. For the penetration assay, triplicates were used for acyclovir, lidocaine, and diclofenac each.

### 2.5. Skin Explants

Two human skin explants (Figure 2A) from one donor undergoing plastic surgery were used with consent for the ex vivo studies after obtaining approval from the local ethics committee of the Medical University of Graz (28-151 ex 15/16 of 22 December 2019). Skin explants were cut to shape, any excess fat was removed around the edges, and explants were affixed onto a covered Styrofoam^TM^ board using cannulae.

Explants were prepared similarly to the FTSE setup, by implanting OFM probes, attaching O-shaped patches, connecting OFM pumps, tubing, online monitoring systems, and sampling vials. Investigations with skin explants were performed inside the ex vivo incubator. Online monitoring of the skin explants was performed with one sample. For the penetration assay, four replicate sites were used for acyclovir, lidocaine, and diclofenac each.

### 2.6. Penetration Tests

Commercially available creams were selected to test penetration with the active pharmaceutical ingredients (API) acyclovir, lidocaine, and diclofenac, which differ in their ability to penetrate dermal tissue (Table 1). Creams were first applied inside the O-shaped application sites using pipettes (Eppendorf, Hamburg, Germany, and Gilson S.A.S., Villiers-le-Bel, France) and then spread evenly using finger cots (Henry Schein, e.g., Latex finger cots). For precise dosing of creams, preliminary tests were carried out considering the residual amount of cream remaining in pipette tips and adhering to the finger cots. Application sites were not occluded throughout the experiments. Applied creams and amounts are depicted in Table 1. FTSE penetration was monitored by measuring API concentrations in the culture medium and in OFM samples from FTSE tissue. Lidocaine and diclofenac samples were collected hourly for 12 h after application of the creams. As acyclovir had previously demonstrated a much slower penetration in preliminary experiments, samples were collected for 24 h after application.

### 2.7. Analyses of Active Ingredients by HPLC-MS/MS

All OFM samples were stored at −80 °C until further processing. **Acyclovir** was analyzed as described before [40]. For **diclofenac** we diluted OFM samples, calibrators, and quality controls. Samples were then mixed with equal volumes of internal standard solution [diclofenac-13C6 (Sigma-Aldrich, St. Louis, MO, USA) in water] and 5% (*v*/*v*) formic acid in water and applied to a preconditioned solid-phase microelution plate (SPE, Strata™-X Polymeric Reversed, microelution 96-well plate, 2 mg/well, Phenomenex Inc., Torrance, CA, USA). Loaded samples were washed with water and 50% (*v*/*v*) methanol in water prior to elution with 1% (*v*/*v*) formic acid in acetonitrile. Eluates were dried with compressed air and reconstituted in 30% (*v*/*v*) acetonitrile in water prior to quantitative analyses with a UHPLC/QqQ system [UHPLC 1290 Infinity II/QqQ 6495B (Agilent Technologies, Santa Clara, CA, USA)]. Isocratic separation of diclofenac (50% (*v*/*v*) acetonitrile in water) was achieved with an ACQUITY BEH C8 reversed-phase pre and analytical column [2.1 × 30 mm, 1.7 μm (Waters Corporation, Milford, CT, USA)] at a flow rate of 400 µL/min and 35 °C. Negative electrospray ionization was used prior to MS detection (diclofenac MS/MS transition 294 > 250, internal standard transition 300 > 256). The analytical range was from 0.9 to 186.0 ng/mL. **For lidocaine**, we diluted OFM samples, calibrators, and quality controls in the perfusate. Samples were then mixed with equal volumes of internal standard solution [lidocaine-d10 (Toronto Research Chemicals, Toronto, ON, Canada) in 1.5% (*v*/*v*) formic acid in water] and applied to a preconditioned Strata™-X SPE plate (see above). Loaded samples were washed with water prior to elution with acetonitrile. Eluates were dried with compressed air and reconstituted in 5% (*v*/*v*) acetonitrile in water prior to quantitative analysis using HPLC coupled to a high-resolution mass spectrometer [HPLC Ultimate 3000/HRMS LTQ Orbitrap XL (Thermo Fisher Scientific, Waltham, MA, USA)]. Chromatographic separation of lidocaine was achieved using an Atlantis T3 reversed-phase column [2.1 × 150 mm, 3 µm, (Waters Corporation, Milford, MA, USA)] and gradient elution with two mobile phases [0.1% (*v*/*v*) formic acid in water (mobile phase A); 0.1% (*v*/*v*) formic acid in acetonitrile (mobile phase B)] at a flow rate of 300 µL/min and a column temperature of 25 °C within 7 min run time. The multistep gradient started with 5% mobile phase B, went up to 50% mobile phase B at 1.5 min and up to 90% at 2.0 min, and kept at 90% until 4 min. Positive-heated electrospray ionization was used prior to full scan mass detection from *m*/*z* 215 to 255. The analytical range was from 1 to 1000 ng/mL.

### 2.8. Statistical Analysis

Statistical analysis was performed using Microsoft Excel (version 2016, Microsoft Corporation, Albuquerque, NM, USA). Data are represented as the mean ± standard deviation (SD). API concentrations in FTSE and skin explants were compared by calculating the area under the curve AUC. Two group comparisons were performed with a non-parametric Mann–Whitney U test. Significance was accepted at *p* ≤ 0.01.

## 3. Results

### 3.1. Online Monitoring of Critical Tissue Parameters

We used OFM to monitor culture parameters in an in vitro FTSE and in ex vivo skin explants both situated in an ex vivo incubator under controlled conditions (Figure 1).

OFM probes were perfused with the perfusate (pump head 1) which equilibrates with the ISF of the surrounding tissue and is withdrawn by the pump head 2 (OFM sample). To measure critical tissue parameters directly in tissue rather than in the culture medium, OFM samples were collected at the outlet of the OFM probe which was inserted directly in either FTSE or skin explants (Figure 3A). Consequently, concentrations of culture parameters (i.e., oxygen, pH) and metabolites (i.e., glucose, lactate) found in the OFM sample are more representative of the state of the tissue than those found in the culture medium. Furthermore, the integration of a custom-made microfluidic flow-through cell and commercial sensors allowed the online measurement of oxygen, pH (Figure 3B), glucose, and lactate. FTSE and ex vivo skin explants were cultured in the incubator for up to 24 h while monitoring the mentioned parameters to assess whether the tissues remained viable.

**Oxygen levels** in the culture medium remained relatively constant at 190.7 ± 2.7 hPa, corresponding to the oxygen concentration of the ambient air (Figure 4A). In contrast, the oxygen concentration measured in the FTSE samples was lower at 71.6 ± 36.8 hPa. No oxygen was detected in skin explants.

**pH levels** in the culture medium increased from pH 7.75 to 9.25, in contrast to levels measured in skin explant and FTSE samples, which were relatively constant at pH 7.5 to 8.0. In the first hours, the pH levels of the skin explants were similar to FTSE pH levels but declined slightly over time (pH = 7.60) (Figure 4B).

The **glucose level** in the culture medium remained relatively constant at 328.8 ± 33.3 mg/dl. In contrast, glucose levels in the FTSE samples were around 171.5 ± 26.9 mg/dL, slightly decreasing over time (Figure 4C). No glucose was found in the skin explant.

The **lactate level** measured in the culture medium increased linearly from 0 to 30 mg/dL. Lactate levels measured in FTSE samples showed a similar trend between hours 10 and 18, with levels around 60 mg/dL. Subsequently, the lactate levels in FTSE dropped to lower than starting lactate levels. Levels of lactate found in skin explant samples were around 2 times lower than those found in FTSE (Figure 4D).

### 3.2. Penetration Tests in the FTSE

The experimental setup (Figure 1 and Figure 3) was used to assess the penetration of acyclovir, diclofenac, and lidocaine into FTSE. The implanted OFM probe allowed concentration measurements of the different APIs directly in FTSE tissue rather than only in the culture medium. In all penetration tests, the API levels in the OFM samples from FTSE tissue were significantly higher relative to the culture medium (Figure 5, all *p*-values ≤ 0.01). In general, API concentrations accumulated in the culture medium, whereas levels in FTSE tissue showed concentration maxima at different time points and then declined due to further penetration into the culture medium. For acyclovir, the maximum concentration in FTSE tissue was observed 6–12 h after application (Figure 5D), whereas it was not clear for diclofenac if the major peak occurred within the 12 h of sampling or the concentration was still increasing (Figure 5E). The maximum concentration for lidocaine was observed after 3 h (Figure 5F).

When comparing FTSE tissue to explanted skin, all OFM samples from FTSE tissue had higher API concentrations of acyclovir, diclofenac, and lidocaine relative to OFM samples from explanted skin (Figure 6A–C) at any sampling time. Total API concentrations (area under the curve) in FTSE tissue were also found to be significantly higher relative to skin explants (Figure 6D).

## 4. Discussion

In this study, we used OFM for continuous monitoring of culture parameters in the dermis of an FTSE. We also compared FTSE data to samples collected from the culture medium of the FTSE and to data from skin explants. Furthermore, we performed a penetration assay with acyclovir, diclofenac, and lidocaine and used OFM to monitor API concentrations in FTSE dermis and human skin explants relative to the culture medium.

Standard cell culture parameters such as oxygen, pH glucose, and lactate are often neglected when using in vitro skin equivalents or skin explants for penetration assays. However, it is clear that shifts in the concentration of these parameters can have detrimental effects on cellular behavior. For example, a hypoxic culture environment has been shown to increase the lipid barrier formation of customized FTSE [41].

We have demonstrated before that the actual values of these parameters in 3D-cultured in vitro equivalents differ significantly from the values measured in the culture medium [26]. Knowing the actual parameters in the ISF rather than the culture medium can help to (i) better control the culture process, (ii) improve culture conditions, and finally, (iii) develop better in vitro equivalents. To the best of our knowledge, these parameters have not yet been measured directly in the dermis of skin equivalents or skin explants. The current study revealed significant differences when comparing culture parameters in the dermal ISF of the FTSE to the same parameters measured in the culture medium. We found an increased pH in the culture medium, which can be explained by CO_2_ transfer from the culture medium into ambient air, causing alkalization. In contrast, the pH in the FTSE samples remained relatively stable, which could be attributed to the production of lactate of the living cells in the FTSE to some extent. Especially oxygen and glucose supplies were significantly lower than indicated by measuring the culture medium samples. As expected, oxygen and glucose levels in the skin explants were completely depleted, indicating cell death and tissue degradation. In the skin explant samples, no glucose was detected which can be explained by the fact that the skin explant was not supplied with fresh nutrients and the glucose was already consumed at the time when the measurements were started. Living cells must be within 200 µm of blood supply to be properly supplied with oxygen and nutrients [42]. The extremely low oxygen and glucose concentrations in the explanted tissue can be explained by the missing blood recirculation. In contrast, oxygen and glucose levels remained stable for 24 h in the FTSE. Lactate levels dropped considerably in FTSE towards the end of the sampling time which could be associated with sensitivity loss of the lactate sensors.

A recent review on alternatives to biological skin in permeation studies came to the conclusion that a huge problem is that skin equivalents or surrogates, such as skin-like membranes, fail to estimate the amount of drugs penetrated into different skin layers [43]. By using OFM, we were able to measure the actual concentration of APIs in the dermis of an FTSE. We found that API concentrations were significantly higher compared to the corresponding cell culture medium. These findings indicate that cells in the tissue are exposed to much higher concentrations of the APIs than estimated by measurement of the culture medium only. As expected (Table 1), the penetration ranged from slow (acyclovir) to medium (diclofenac) to fast (lidocaine) for FTSE. For skin explants, acyclovir showed a better penetration than expected possibly because acyclovir has been described to be able to use follicular pathways to pass through the stratum corneum [44,45].

Penetration tests also showed higher API concentrations in FTSE compared to skin explant samples, which indicates that the barrier function of the FTSE is not yet comparable to that of skin explants. This observation has also been frequently made with non-vascularized in vitro skin models [43,46]. For three-layered human skin equivalents that include the hypodermis [47,48,49], OFM technology offers spatiotemporal measurement of various substances in different depths and layers of the skin equivalents. OFM data can lead to a better understanding not only of the penetration speed but also the residence time or accumulation of substances in the different layers. Together with advanced skin equivalents with barrier functions comparable to in vivo skin, OFM technology enables a more accurate characterization of pharmacokinetics in the future.

## 5. Conclusions

In conclusion, the presented advanced approach for online monitoring of culture parameters allows an accurate characterization of in vitro skin equivalents and skin explants. This enables better control of in vitro cultures and the development of more physiological skin equivalents. Furthermore, API monitoring in the dermis of the FTSE and skin explants during penetration assays can determine the actual concentration of substances that cells in the tissue are exposed to. This might enable a more accurate characterization of pharmacological penetration assays in skin equivalents to reduce or avoid animal testing.

## Figures and Tables

**Figure 1 pharmaceutics-14-01436-f001:**
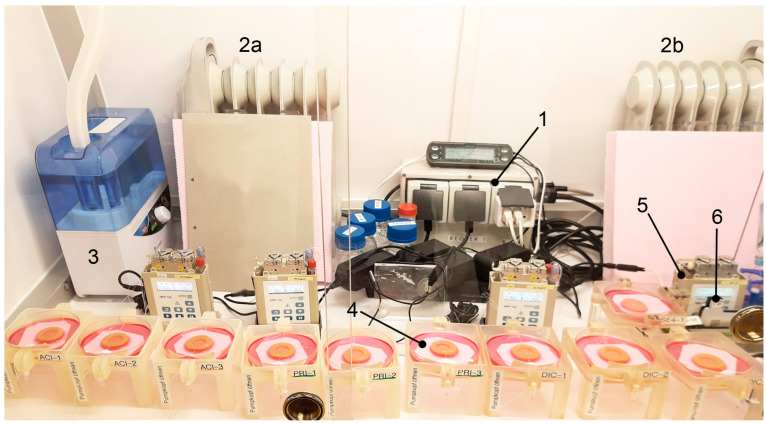
General setup of the ex vivo incubator with (1) climate control unit, (2a and 2b) heater, (3) humidifier (second humidifier on right not shown), (4) sampling units with FTSEs, (5) OFM pumps, and (6) online monitoring flow-through cells.

**Figure 2 pharmaceutics-14-01436-f002:**
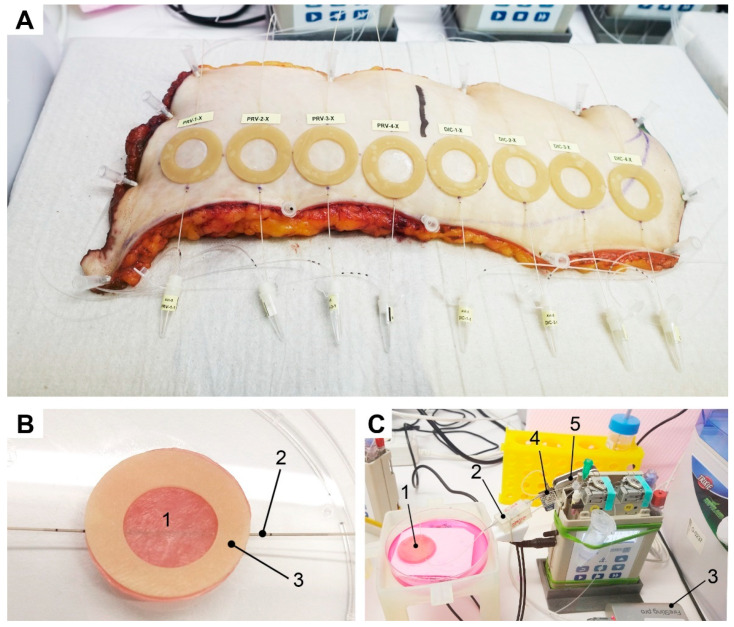
(**A**) Human skin explants with implanted OFM probes, O-shaped patches, and sampling vials. (**B**) Detailed setup of (1) FTSE with (2) an implanted OFM probe and (3) an O-shaped patch to avoid cross-contamination and define the size of the application area. (**C**) Setup of online monitoring (1) FTSE, (2) lead flow-through cell for oxygen and pH, (3) multi-analyte meter FireSting^®^-PRO, (4) second flow-through cell for glucose and lactate, and (5) biosensor measuring instrumentation (Six, Jobst Technologies).

**Figure 3 pharmaceutics-14-01436-f003:**
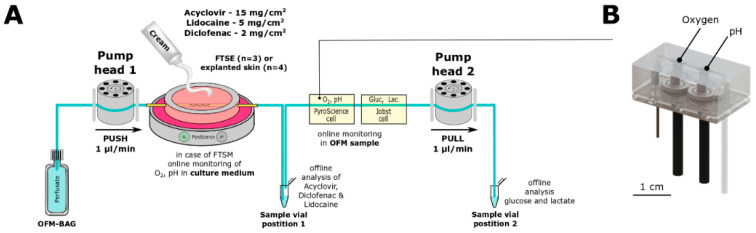
(**A**) Setup for penetration tests: Perfusate is pumped from OFM bag through OFM probe (yellow) implanted in FTSE with topically applied cream and withdrawn by pump head 2. OFM samples for offline analysis of acyclovir, lidocaine, and diclofenac concentrations were collected at position 1. OFM samples for the offline analysis of glucose and lactate were collected at position 2. (**B**) Custom-made microfluidic flow-through cells for online monitoring of oxygen and pH: connection with tubing and optical fibers for data read out.

**Figure 4 pharmaceutics-14-01436-f004:**
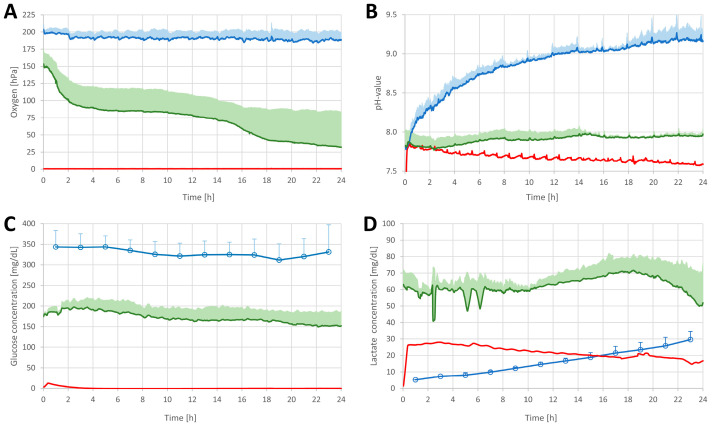
Tissue parameters measured online in FTSE (green line) and skin explants (red line) and culture medium (blue line for oxygen (**A**) and pH (**B**)) and offline in culture medium (blue dots for glucose (**C**) and lactate (**D**)). Data represent the average of n = 3 (for readability only upward SD is shown except for skin explants where only one sample was measured).

**Figure 5 pharmaceutics-14-01436-f005:**
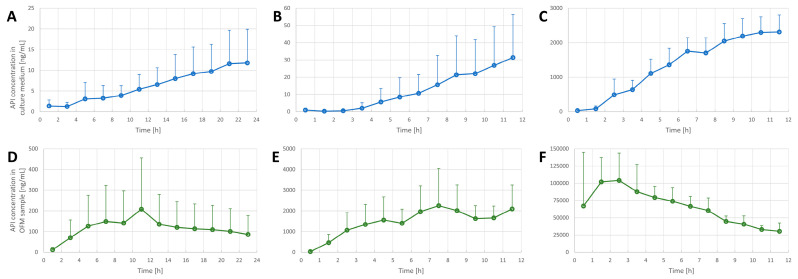
Penetration tests in FTSE (n = 3) using acyclovir (**A**,**D**), diclofenac (**B**,**E**) and lidocaine (**C**,**F**). API concentrations were determined in FTSE culture medium (top row (**A**–**C**)) and in FTSE tissue by OFM sampling (bottom row (**D**–**F**)). Data are shown as the mean and SD (only upward error bars are shown for readability).

**Figure 6 pharmaceutics-14-01436-f006:**
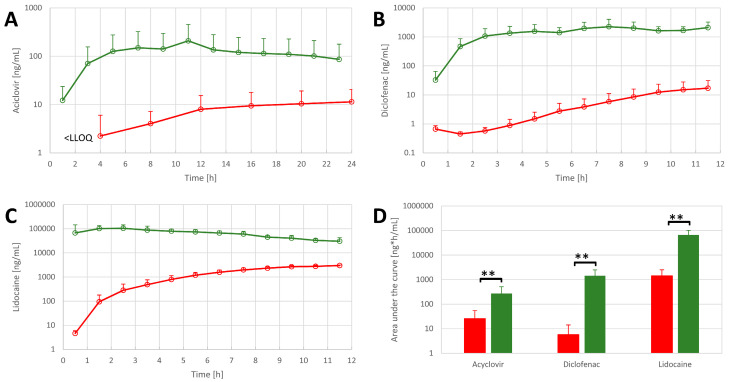
API concentrations in OFM samples from FTSE (n = 3, green) and skin explants (n = 4, red) for (**A**) acyclovir, (**B**) diclofenac, and (**C**) lidocaine. (**D**) Comparison of the total concentration of APIs in OFM samples in FTSE (green bars) and skin explants (red bars). Data are shown as the mean and SD (only upward error bars are shown for readability). ** *p* ≤ 0.01.

**Table 1 pharmaceutics-14-01436-t001:** List of applied creams with their active ingredients and dosing.

Active Ingredients	Registered Trade Name (Manufacturer, City, Country)	Dose	No. Application Sites Explants/FTSE	Penetration Behavior
Acyclovir	ACICLOVIR Creme (1A Pharma GmbH, Vienna, Austria)	15 mg/cm^2^	4/3	Weak
Diclofenac	Voltaren Arthritis Pain Gel 1% (GlaxoSmithKline, Philadelphia, USA)	2 mg/cm^2^	4/3	Medium
Lidocaine	EMLA Cream 5% (AstraZeneca GmbH, Wedel, Germany)	5 mg/cm^2^	4/3	Good

## Data Availability

The data presented in this study are available on request from the corresponding author.

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
