# Peer review of "Advanced Online Monitoring of In Vitro Human 3D Full-Thickness Skin Equivalents"

_pharmaceutics, 2022, doi:10.3390/pharmaceutics14071436_

Round 1

Reviewer 1 Report

Dear Authors, 

The manuscript entitled "Advanced Online Monitoring of in vitro Human 3D Full Thickness Skin Equivalents" addresses the evaluation of the critical parameters in cell cultures (oxygen, pH, glucose, and lactate) in skin equivalents which also compares to the skin explants. To monitor those parameters, the study has been conducted with the Open Flow Microperfusion technique under strictly controlled conditions in the incubator. 

The manuscript is well-written and well-organized, the methodology is described precisely and the results are clearly presented and discussed. The references included in the introduction are relevant and cover a wide period of time, particularly considering the latest research on the topic and related fields. This is nice work that shed new light on the drug penetration and permeation assays. 

I just have one comment: 

The title in axes in figures 3, 4 and 5 are a little blur, especially in figure 4. I suggest increasing the quality of the plots. 

Kind regards,

Author Response

Question 1:

The title in axes in figures 3, 4 and 5 are a little blur, especially in figure 4. I suggest increasing the quality of the plots.

Answer 1:

Dear reviewer 1, thank you for your positive feedback. As you suggested we increased the quality of the plots for figure 3, 4 and 5.

Reviewer 2 Report

The manuscript is dealing with the development of an ex vivo incubator combining OFM sampling technology with a biosensor platform for the real-time monitoring of culture parameters. In general, the manuscript is written in a clear way and the literature is sufficiently covered. The methods are fairly described but the results’ presentation, as well as the discussion part, have room for improvement.

My recommendation is to consider acceptance of the manuscript for publication in Pharmaceutics after revision based on the following comments.

Detailed comments:

line 110: The suggested ex vivo incubator is described as „mobile“. Please comment how mobile can be a device with a total weight of 80 Kg while having several smaller and bigger components? In addition, what is included in the total mass of 80 Kg? I assume that the laminar box where all the smaller devices are placed is not included in this total mass, isn’t it? The sampling units are quite big taking up big space. Would it be a real suggestion to substitute them with sampling units of smaller size? What are the approximate total dimensions of the incubator?

line 201: Figure 1:

a) Please replace Figure 1A immediately after paragraph 2.1. Ex vivo incubator

b) The indication for the heater (number 2) seems to be placed on something that looks like a heating isolation surface rather than the actual heater which is behind the isolation surface. Is it true? Please comment. In addition, please add the number 2 on the second heater as well. I would suggest further specification like 2a and 2b for the two heaters.

c) The second humidifier is missing from Figure 1. Can you please provide a photo including the second humidifier and specify it like 1a and 1b similarly to the heaters?

d) The indication for the OFM pump (number 5) also seems to be placed on something that looks like an isolation surface like before.

e) It is difficult to imagine what the number 6 is representing. Maybe combine with Figure 2B where the flow-through cells are presented more closely.

f) Are the flow-through cells attached to the OFM pumps?

line 208: The authors are claiming removal of the excess fat, but this is not visible in Figure 1C where the fat is visible under the skin tissue. Can you please further comment and/or rephrase?

lines 321-324: The authors are describing the permeation profile of three permeants but no statistical significance is mentioned for the measurements while the error bars in Figure 4A-F are quite wide. Were there any statistically significant differences between the values or the described comments are based only on the noticed trend? (The same for the results presented in Figure 5)

line 374-375: In the text, it is mentioned that „the penetration ranged from slow to fast with acyclovir < diclofenac < lidocaine and skin explants < FTSE“. Figure 5D is showing that the order acyclovir < diclofenac < lidocaine was not the case for the skin explants (red columns). Please comment or rephrase.

Lines 327-329: The authors report that “At the end of the penetration test FTSE samples had higher API concentrations for acyclovir, diclofenac, and lidocaine relative to the culture medium (Figure 5A to Figure 5C).” I suppose that Figure 4 shows the comparison between FTSE samples and the culture medium while Figure 5 is showing the comparison between FTSE samples and skin explants. Please correct the text or further specify.

Figure 4 and Figure 5: The results presented in Figures 4D-4F for the concentration of active substances in FTSE are the same as those presented in Figure 5A-5C (green lines) but with a different scaling on Y-axis. Why was a different scaling used? Couldn’t you combine all results in the same Figure for direct comparison?

Author Response

Author's Reply to the Review Report (Reviewer 2):

The manuscript is dealing with the development of an ex vivo incubator combining OFM sampling technology with a biosensor platform for the real-time monitoring of culture parameters. In general, the manuscript is written in a clear way and the literature is sufficiently covered. The methods are fairly described but the results’ presentation, as well as the discussion part, have room for improvement.

My recommendation is to consider acceptance of the manuscript for publication in Pharmaceutics after revision based on the following comments.

Answer:

Dear reviewer 2, thank you very much for your valuable comments. We have addressed all of your suggestions to increase the quality of the manuscript.

Detailed comments:

Question 1: line 110: The suggested ex vivo incubator is described as „mobile“. Please comment how mobile can be a device with a total weight of 80 Kg while having several smaller and bigger components? In addition, what is included in the total mass of 80 Kg? I assume that the laminar box where all the smaller devices are placed is not included in this total mass, isn’t it? The sampling units are quite big taking up big space. Would it be a real suggestion to substitute them with sampling units of smaller size? What are the approximate total dimensions of the incubator?

Answer 1:

Our aim for building this ex vivo incubator was to have a device that can be moved between various laboratories of different working groups using a trolley. Currently, no such device is available on the market. All components can be removed and the laminar box can be placed on a trolley by two operators and moved from one lab to the other. The dimensions of the ex vivo incubator are 140 x 86 x 75 cm (L x B x H).

The sampling units are relatively large because they also hold the Petri dishes with the culture medium and the FTSE in place. To enable correct sampling the distance between FTSE and OFM sample vial is predefined to avoid unintentional pressure differences. Furthermore, the sampling units hold the sampling vials into position and allows a comfortable and safe exchange of the vials. Therefore, our design offers minimum space requirements with maximum performance.

To explain this better, we added the following information to the manuscript in section Material & Methods (lines 116-118): The dimensions of the ex vivo incubator were 140 x 86 x 75 cm (L x B x H) with a maximum weight of the laminar box without components of 80 kg to keep it mobile using a trolley.

Question 2a

line 201: Figure 1:

Please replace Figure 1A immediately after paragraph 2.1. Ex vivo incubator

Answer: 2a

We have now placed Figure 1 immediately after paragraph 2.1. Ex vivo incubator

Question 2b

The indication for the heater (number 2) seems to be placed on something that looks like a heating isolation surface rather than the actual heater which is behind the isolation surface. Is it true? Please comment. In addition, please add the number 2 on the second heater as well. I would suggest further specification like 2a and 2b for the two heaters.

The second humidifier is missing from Figure 1. Can you please provide a photo including the second humidifier and specify it like 1a and 1b similarly to the heaters?

The indication for the OFM pump (number 5) also seems to be placed on something that looks like an isolation surface like before.

It is difficult to imagine what the number 6 is representing. Maybe combine with Figure 2B where the flow-through cells are presented more closely.

Are the flow-through cells attached to the OFM pumps?

Answer  2b

Thank you for your interest in the experimental setup. We have revised the picture and added lines to clearly identify the different components. For the heaters, we added additional information (2a, 2b) to the figure and caption. Unfortunately, we do not have a full photo of the incubator with the second humidifier but we added the information in the caption.

We have added the following sentence (lines 123-125) to the manuscript to clarify: Both heaters were covered with StyrofoamTM to prevent direct heat radiation and thus prevent the skin/ FTSE from drying out.

We have updated Figure 1 and the respective caption according to the reviewer’s comments: 

(refer to manuscript)

Figure 1: General setup of the ex vivo incubator with (1) climate control unit, (2a & 2b) heater, (3) humidifier (second humidifier on right not shown), (4) sampling units with FTSEs, (5) OFM pumps and (6) online monitoring flow through cells.

In addition, we have added a photo showing more details of the flow-through cells which are not visible in Figure 1. We have combined former Fig. 1B/1C and the new photo into a new Figure 2.

(refer to manuscript)

Figure 2: (A) Human skin explants with implanted OFM probes, O-shaped patches and sampling vials. (B) Detailed setup of (1) FTSE with (2) an implanted OFM probe and (3) an O-shaped patch to avoid cross contamination and define the size of the application area. (C) Setup of online monitoring (1) FTSE, (2) lead flow-through cell for oxygen and pH, (3) multi analyte meter FireSting®-PRO,  (4) second flow-through cell for glucose and lactate, and (5) biosensor measuring instrumentation (Six, Jobst Technologies)

Flow-through cells are placed directly after the outflow of the OFM probes. The OFM sample is then withdrawn with the pump head 2 and thus transported through the flow-through cells. Please refer to new Figure 2C and Figure 3.

Question 3:

line 208: The authors are claiming removal of the excess fat, but this is not visible in Figure 1C where the fat is visible under the skin tissue. Can you please further comment and/or rephrase?

Answer 3:

Immediately after surgery, skin explants had a much more excessive amount of fat. This excessive fat was removed and skin explants were cut into shape to enable pinning on the StyrofoamTM board.

We changed line 216-217 of the manuscript accordingly: Skin explants were cut to shape, any excess fat was removed around the edges, and explants were affixed onto a covered StyrofoamTM board using cannulae.

Question 4:

lines 321-324: The authors are describing the permeation profile of three permeants but no statistical significance is mentioned for the measurements while the error bars in Figure 4A-F are quite wide. Were there any statistically significant differences between the values or the described comments are based only on the noticed trend? (The same for the results presented in Figure 5)

Answer 4:

Thank you for your suggestion. We have calculated the p-values using a non-parametric Mann–Whitney U test and all differences were significant with p-values ≤ 0.01. We have added this information to the results section (line 332-334) and in the caption of Figure 5 (former Figure 4). For Figure 6 (former Figure 5) the summarized results incl. statistics are presented in Figure 6D.

Question 5:

line 374-375: In the text, it is mentioned that „the penetration ranged from slow to fast with acyclovir < diclofenac < lidocaine and skin explants < FTSE“. Figure 5D is showing that the order acyclovir < diclofenac < lidocaine was not the case for the skin explants (red columns). Please comment or rephrase.

Answer 5:

Thank you for reading our manuscript in such detail. We agree that the explanation was confusing and rephrased the section in the discussion (382-385) and added two relevant citations: As expected (Table 1) the penetration ranged from slow (acyclovir) to medium (diclofenac) to fast (lidocaine) for FTSE. For skin explants, acyclovir showed a better penetration than expected possibly because acyclovir has been described to be able to use follicular pathways to pass through the stratum corneum (Ogiso, T., Shiraki, T., Okajima, K., Tanino, T., Iwaki, M., Wada, T., 2002. Transfollicular drug delivery: Penetra-tion of drugs through human scalp skin and comparison of penetration between scalp and abdominal skins in vitro. J. Drug Target. 10, 369–378. https://doi.org/10.1080/1061186021000001814

Frum, Y., Eccleston, G.M., Meidan, V.M., 2008. Factors influencing hydrocortisone permeation into human hair follicles: Use of the skin sandwich system. Int. J. Pharm. 358, 144–150. https://doi.org/10.1016/j.ijpharm.2008.02.030)

Question 6:

Lines 327-329: The authors report that “At the end of the penetration test FTSE samples had higher API concentrations for acyclovir, diclofenac, and lidocaine relative to the culture medium (Figure 5A to Figure 5C).” I suppose that Figure 4 shows the comparison between FTSE samples and the culture medium while Figure 5 is showing the comparison between FTSE samples and skin explants. Please correct the text or further specify.

Answer 6:

We previously used ambiguous wording to describe OFM tissue samples. We have now clarified this wording throughout the manuscript to clearly describe OFM samples as either FTSE OFM sample or skin explant OFM sample. Figure 5 (used to be Figure 4) shows the comparison of the API concentrations in FTSE culture medium (top, blue) and FTSE tissue (OFM samples – bottom, green). We have corrected the caption of the figure, calculated the significance values and added them in the text (line332-334): In all penetration tests, the API levels in the OFM samples from FTSE tissue were significantly higher relative to the culture medium (Figure 5, all p-values ≤ 0.01).

It is correct that Figure 6 (used to be Figure 5) is showing the comparison of OFM samples in FTSE and skin explants. We corrected the text of the manuscript (line 334-338): When comparing FTSE tissue to explanted skin, all OFM samples from FTSE tissue had higher API concentrations of acyclovir, diclofenac, and lidocaine relative to OFM samples from explanted skin (Figure 6A -6C) at any sampling time. Total API concentrations (area under the curve) in FTSE tissue were also found to be significantly higher relative to skin explants (Figure 6D).

Question 7:

Figure 4 and Figure 5: The results presented in Figures 4D-4F for the concentration of active substances in FTSE are the same as those presented in Figure 5A-5C (green lines) but with a different scaling on Y-axis. Why was a different scaling used? Couldn’t you combine all results in the same Figure for direct comparison?

Answer 7:

We used a different scaling for the graphs to be able to show both lines in one graph. Without the logarithmic scale the red lines in Figure 6 would disappear on the x-axis due to the large penetration differences between FTSE and explanted skin.

We did not combine Figure 5 and Figure 6 for readability reasons. On the one hand, Figure 5 compares the API values in FTSE in the culture medium and in the OFM samples. Here we do not use a logarithmic scale.

On the other hand, we use a logarithmic scale in Figure 6 to compare the values of OFM samples in FTSE and explanted skin, because the difference in penetration is that big.

Reviewer 3 Report

3.2? Missing experimental data!

Author Response

dear reviewer 3

Thank you very much for the valuable input to improve the manuscript.

We are working on it to integrate your comments and suggestions.

Unfortunately we are unable to interpret the suggestion of reviewer 3:

" 3.2? Missing experimental data!"

Would you be so kind and give us more information what is meant by this comment?

Br roland

Round 2

Reviewer 3 Report

2.5: "Data are represented as the mean ± standard deviation (SD).". In the paper, the data has only positive deviation.

Author Response

Author's Reply to the Review Report (Reviewer 3):

Question 1

The author wrote 3.1 and 3.3, and 3.2 is missing. I thought that the author had content but not written.

Answer 1:

Thank you for paying close attention to our manuscript. There has been an error in the heading numbers which we have now corrected. There is no experimental data missing. We have corrected the numbering throughout the manuscript.

Question 2:

2.5: "Data are represented as the mean ± standard deviation (SD).

In the paper, the data has only positive deviation.

Answer 1:

We are only presenting the upward error bars to ensure better readability in the graphs and have changed all figure captions accordingly. The description in 2.9 (used to be 2.5) refers also to the text sections where we are presenting the data as mean ± standard deviation (SD) e.g. lines 301-310.

best regards Roland